# *I Can Tell What I Am Doing:* Toward Real-World Natural Language Grounding of Robot Experiences

**Zihan Wang, Brian Liang, Varad Dhat, Zander Brumbaugh**
**Nick Walker, Ranjay Krishna, Maya Cakmak**
Paul G. Allen School of Computer Science & Engineering
University of Washington
`avinwang@cs.washington.edu`

**Abstract:** Understanding robot behaviors and experiences through natural language is crucial for developing intelligent and transparent robotic systems. Recent advancement in large language models (LLMs) makes it possible to translate complex, multi-modal robotic experiences into coherent, human-readable narratives. However, grounding real-world robot experiences into natural language is challenging due to multi-modal nature of data, differing sample rates, and data volume. We introduce RONAR, an LLM-based system that generates natural language narrations from robot experiences, aiding in behavior announcement, failure analysis, and human-assisted failure recovery. Evaluated across various scenarios, RONAR outperforms state-of-the-art methods and improves failure recovery efficiency. Our contributions include a multi-modal framework for robot experience narration, a comprehensive real-robot dataset, and empirical evidence of RONAR's effectiveness in enhancing user experience in system transparency and failure analysis. Supplementary material and videos can be found on the project website here.

**Keywords:** Large Language Model, Explainable AI, Failure Analysis

## 1 Introduction

Natural language is important for understanding robots' behaviors and experiences [1], which is essential for creating more intelligent and transparent robotic systems. Some previous works have attempted to make robotic systems more transparent using language [2, 3], but these efforts are either limited to specific domains or dependent on task setups. With the rapid progress of large language models, there is a growing body of work in robotics [4, 5, 6], including planning [7, 8, 9], decision-making [10, 11] and robot learning [12, 13, 14, 15]. The natural language interface and commonsense reasoning capabilities inherent to large language models (LLMs) make it feasible to ground general real-world robot experiences within the domain of natural language. Providing grounded, real-world robot experiences can greatly enhance system transparency, leading to improved safety and user satisfaction in applications such as assistive feeding, home robotics, and self-driving vehicles.

Grounding real-world robot experiences into natural language presents three main challenges [16]. First, robot data is multi-modal, making it difficult to process and integrate. Mobile manipulators produce RGB images from various viewing angles, point clouds, audio recordings and more sensor data. These disparate data formats and semantics complicate the processing and integration of multi-modal inputs. Secondly, robot data has different sample rates, making alignment difficult. Visual data from cameras might be captured at a different frequency than data from force sensors or joint encoders. This discrepancy complicates synchronizing data streams to create a coherent representation of the robot's experience. Lastly, robot data is voluminous, making real-time narration challenging. Processing this data in real-time to produce meaningful and accurate natural language

8th Conference on Robot Learning (CoRL 2024), Munich, Germany.

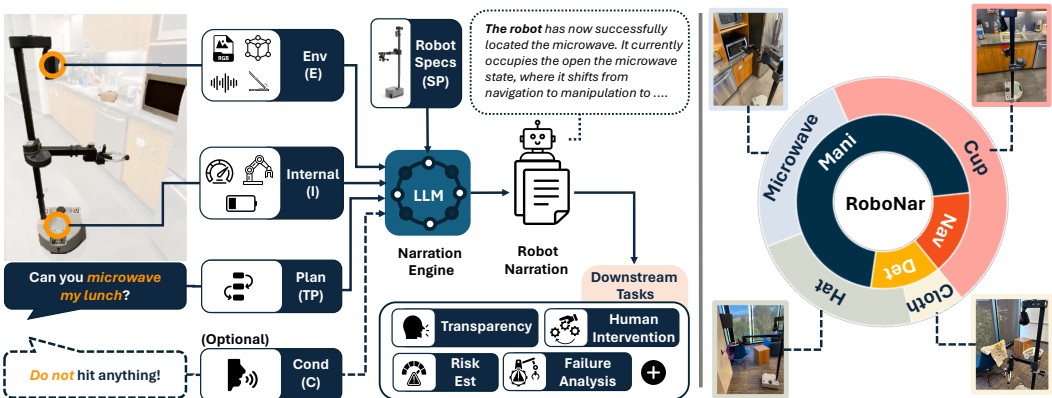

Figure 1: **Left:** Our framework for real-world robot narration, RONAR. It takes in four categories of dynamic inputs and one static input: multimodal environmental observations (E), robot internal states (I), task planner (TP), and specified conditions (C), along with robot specifications (SP). RONAR then uses its LLM-based narration engine to process these inputs and generate narrations based on the specified narration mode. The generated narration can be used to address downstream narration-related tasks. **Right:** The RoboNar dataset. It includes four daily housekeeping tasks with real failure cases, containing ground truth failure explanations and recovery descriptions labeled by human experts.

narratives requires substantial computational resources. Additionally, the system must filter and prioritize relevant information to avoid overwhelming the user with unnecessary details.

In this paper, we introduce a LLM-based real-world robot narration system, RONAR, which can generate robot narrations based on the experiences and to resolve downstream tasks, including behaviour announcement, risk estimation, failure analysis, and human intervention. We considered users with different use cases and levels of expertise, ranging from no prior experience with robots to expert-level familiarity. These scenarios incorporated different narration modes with diverse narration strategies and abstraction levels. We created a real-robot dataset encompassing four daily housekeeping tasks with domain randomization by using Stretch, a single arm mobile robot. Finally, we conducted experiments involving numerical comparisons and user studies to assess the quality and effectiveness of the narrations. The key contributions of this paper are:

- A modular multi-modal framework to ground robot experiences into natural language in the real world using LLMs;

- Real home-robot dataset that includes four common home tasks with realistic failure cases across navigation, manipulation, and detection;

- Empirical experiments on failure analysis tasks using narrations from both robot system perspective and user perspective which demonstrates that our system outperforms state-of-the-art methods and can improve users' failure recovery efficiency.

## 2   Related Works

**Robotics and LLMs.** LLMs and VLMs can be used in almost every part of robot development [5, 6]. In perception, vision-language model and vision-language-action models have demonstrated they can significantly enhance the generalization capabilities of robots [14, 17, 18, 19, 20]. In decision-making, LLMs have been used to make planning and execution more flexible and context-sensitive [7, 21, 22, 23, 24, 25, 26]. In control, language-conditioned policies and transformer-based robot control have been widely studied in recent research [7, 27, 28, 29, 14, 30, 31, 32, 13]. Finally, LLMs can significantly improve both robot-environment and robot-human interactions [33, 34]. In this work, we focus on using LLMs to enable natural language grounding of robot experiences and applying the grounded experiences to resolve downstream real-world robot tasks.

**Scene and Action Understanding for Robots.** Scene understanding involves recognizing objects, their relationships, and the context within a scene, while action understanding requires interpreting the robot's actions, understanding the outcomes, and planning future actions accordingly. Although dense video captioning and scene-graph generation have been extensively studied in computer vision [35, 36, 37, 38, 39, 40], these models cannot be directly transferred to robots due to distribution mismatches. With the rise of embodied AI and LLMs, new approaches for scene representation and understanding in robots have emerged [11, 41, 42, 43, 44, 45, 46]. Scene and action understanding must work together to solve robotic tasks, such as failure explanation [26, 47, 48, 49, 50, 51], affordance estimation [7, 52], and task execution [53, 54, 55, 56, 57]. We introduce a framework to ground robot experiences with both scene and action understanding. Unlike previous work [26], our framework includes both low-level control and high-level planning actions.

**Robot Transparency.** Transparency, the ability to *see into* a robot's behavior, can help users accept, collaborate with, or debug a robot [58, 59]. Many channels for providing transparency, like gaze or expressive motion, are only appropriate for exposing limited amounts of information [60], and richer alternatives like displays or projections often require additional hardware [61, 62]. Language is a naturalistic means of providing information about a robot's behavior which requires only a speaker or a display, but previous approaches have been constrained to engineered templates, often in conjunction with symbolic models of the behavior [63]. Our framework does not require any engineered templates or symbolic models, which makes it more generalizable and easy to use.

## 3 RONAR: The Real-World Robot Narration Framework

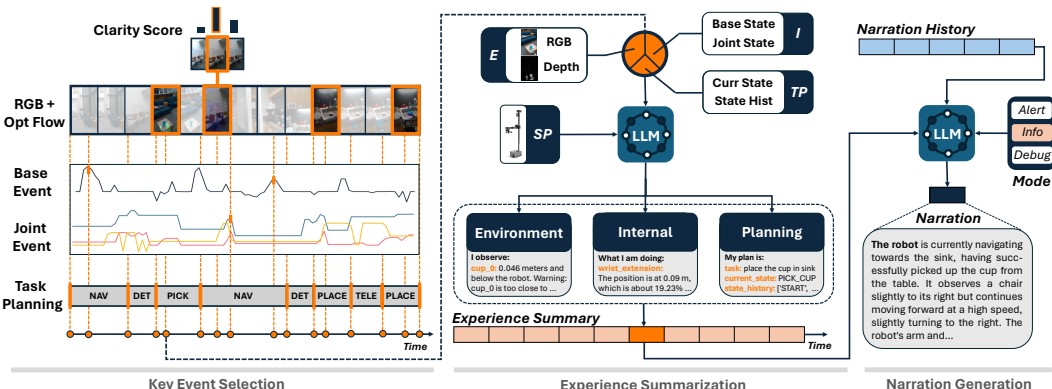

Figure 2: **RONAR:** Our framework for real-world robot narration. It has three parts, which are key frame selection, experience summarization and narration generation. It takes in the raw multimodal robot data stream and outputs text describing past experiences, current observations, and future plans of the robot.

### 3.1 Multimodal Key Event Selection

When executing robot processes, massive amounts of data are streamed at persistent rates across different sensors on the robot, which can create data that is too repetitive and dense to be interpretable for users. Furthermore, the sampling rates of different robot sensors can vary significantly, creating difficulties in aligning information to be processed together. These challenges necessitate procedures for aligning and sampling valuable information from robots. We call these results key events, which are sampled from the raw robot data and used to generate experience summaries in Section 3.2 and narrations in Section 3.3.

**Multi-Sensory Data Alignment.** We split robotic data into three categories: Environment (E), Internal (I) and Task Planning (TP). We sample and align these dense, mixed data by dividing the duration of the data by a single sample rate $s$ for a sequence of frames $f_0, f_1, \ldots, f_n$, such that frame $f_i$ and $f_{i+1}$ are separated by time s. In each frame, we add the robot information across each

considered medium, using the information with the timestamp closest to the frame timestamp. The result of this procedure is a sequence of frames separated by a fixed interval that captures information across robot sensors that may have mixed, high-frequency sampling rates.

**Key Event Selection with Multimodal Inputs.** Key events are selected from the aligned data by heuristically monitoring for interesting information across the different data categories. For environmental data, we compute the optical flow of the RGB images to capture the motion dynamics within the scene, using the running sum of average flow magnitudes as a heuristic for computing changes in perception information sufficient for a key event. For the internal state of the robot, the joint states are used as a heuristic for observing changes in robot motion sufficient for a key event. Observing that changes in both flow magnitudes and joint states are significantly different for when the robot is moving its base, moving its camera, and all other movements, we normalize each of these values to have a mean of zero and standard deviation of one, and track the running positive sum of the normalized values. Once the running sum reaches a set threshold, we note that there should be a key event. We also add a key event each time there is a state change in the task planner, with the reasoning that state transitions are indications of notable events. Details can be found in Appendix.

## 3.2 Experience Summarization

With selected key events, the next step is to ground raw robotic data into experience summaries in natural language. Based on the categories of the robot data, the experience summary is also composed of three components: environment summary, internal summary, and planning summary.

**Environment Summary.** The goal of an environment summary is to ground the observations from the robot into natural language. We use YOLO World [64] to conduct open-world object segmentation of the corresponding RGB images. The detected objects are represented by a bounding box coordinate and unique object id, forming an detected object set, $O_{det}$. Since the real environment is complex, the observation of a scene can be complicated with excessive number of objects in it. Our system leverages depth information to filter out irrelevant objects based on certain distance criteria, $c_d$. The remaining objects forms an object set for the scene,

$$O_s = \{o_s | (o_s \in O_{det}) \cap (d_{o_s} < c_d)\} \tag{1}$$

where $d_{o_s}$ is the distance between object $o_s$ and the the robot. We have a spatial relation set for the objects, which is defined as $P_s = \{$*left to, right to, above, below, in front of, behind*$\}$. The scene graph is a set of object, relation and distance triplets, $R_s \subseteq O_s \times P_s \times D_s$.

**Internal Summary.** The goal of the internal summary is to ground numerical values of part states (e.g. base states, joint states, etc.) to natural language based on the configuration of the robot. Each part of the robot in the configuration has three components: part description, part limit and part type. The robot's configuration is specified as part of the system prompt for the internal summary generation. The system prompt also specified that the internal summary should contain exact names and numerical values of the part states. The final internal summary contains a list of robot parts with exact part names, a detailed description with numerical values, and a grounded explanation that people without robot experiences can understand (see a detailed example in the Appendix).

**Planning Summary.** It is hard to infer and narrate the expected outcomes for every one of the low-level actions contained in the internal summary. Therefore, a planning summary is generated to capture the plan-level status. It summarizes the high-level plan of task execution. It contains the overall task with description, the sequential order of sub-goals, the current sub-goal and a history of sub-goal executions and outcomes (see a detailed example in Appendix). Unlike other methods only using the current sub-goal in their planning summary, one critical change is we also include a history of sub-goal executions and outcomes. It would help identify the correlations between failures across sub-tasks and enable narration and failure analysis for long-horizon tasks.

### 3.3 Narration Generation

The experience summaries ground environmental observations, internal status and task planning of a robot during task execution into detailed natural language. However, not all of the details are useful for humans to understand and react to the robot. We need to abstract the information and only narrate things users care about.

**Narration Mode.** The requirements for narrations can vary significantly depending on the robot's use cases and the user's level of expertise. To meet general narration needs, we have defined three narration modes: 1) *Alert Mode* narrates only the important information which requires the user's attention; 2) *Info Mode* narrates robot experiences in multiple sentences and provide a concise summary of the robot's observations, internal status and planning without any numerical values and part names; and 3) *Debug Mode* incorporates all details of environmental observations, robot internal status and the robot's planning, including numerical values and attribute names. Users can specify the mode of narration as needed and the mode is as an input parameter to the LLMs and controls the properties of generated narrations.

**Progressive Narration Generation.** We consider a generated narration to be good if it has the properties of non-repetition and being smooth. Non-repetition means the narration should not repetitively narrate behaviors which has already been narrated to the user. Being smooth means the transition between narrations should be natural and seamless. We use progressive generation to achive these properties. As shown in Figure 2, consider a robot narration history which contains all the narration instances until key event $t-1$, $N_{t-1} = \{n_0, n_1, ..., n_{t-1}\}$. There is a new key event, $k_t$, has been detected and a experience summary, $s_t$ has been generated. We input both $N_{t-1}$ and $s_t$ with a specified mode $m$ to an LLM to generate the narration, $n_t$, at time $t$, where $n_t = LLM(N_{t-1}, s_t | m)$. Then, we add the narration $n_t$ to the narration history. The most recent generated narration $n_t$ will be shown to user in our user interface.

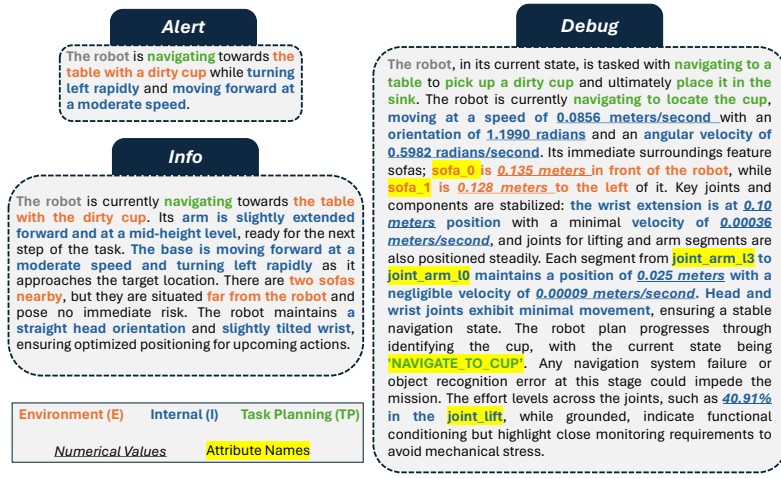

Figure 3: Example of narrations generated by RONAR with different modes.

## 4 Experiments and Results

### 4.1 Experiment Setup

**RoboNar Dataset.** We collect a real-world dataset using a Stretch SE3 robot in a home environment [65]. The details of the home setup can be found in Appendix. We created four real-world housekeeping tasks: pick a dirty cup and put it in the sink, microwave lunch, hang a hat, and collect dirty clothes. We collected data, including RGB-D observations captured by two cameras, an Intel RealSense d435i and d405, joint readings, base readings, state information, and diagnostics. We

also save the processed data, which includes downsampled aligned keyframes. For each demonstration, human experts create ground truth labels for failure timestamps, failure reasons, and recovery instructions. The dataset contains 70 demonstrations and 76 failure cases across navigation, manipulation, and detection. Figure 4 shows a detailed task and failure composition of the dataset.

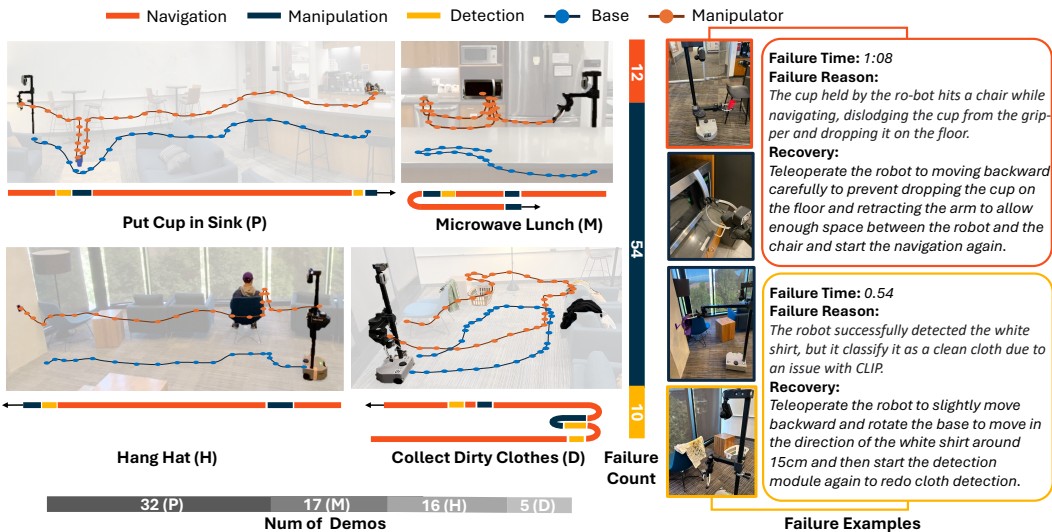

Figure 4: **RoboNar Dataset:** We design four long-horizon tasks for a Stretch robot in a home environment. **Left:** the different tasks with base and manipulator trajectories. It also shows states the robot experiences in each task. **Right:** the number of failure cases under each robot state in the dataset. The pictures are failure cases selected from the dataset and the text are human-expert-provided ground truth labels for the frames.

**Effectiveness of Multimodal Key Event Selection.** We first evaluate the effectiveness of multimodal key event selection in terms of number of captured frames and failure capture rate. We adjust the combinations used of modalities and key event selection threshold to learn the effects and improvements of multi-modal key event selection on sampling efficiency of narration generation. The results are shown in Table 1. More details of experiment setup are given in the Appendix.

| | Thresh | 0 | 5 | 10 | 20 | 40 | 80 | 160 |
|---|---|---|---|---|---|---|---|---|
| **Average Frame Count** | **E** I TP | 1017.97 | 145.38 | 82.81 | 44.63 | 22.97 | 11.44 | 5.59 |
| | E **I** TP | 1017.97 | 121.72 | 68.44 | 36.97 | 19.41 | 9.75 | 4.75 |
| | E I **TP** | 11.94 | 11.94 | 11.94 | 11.94 | 11.94 | 11.94 | 11.94 |
| | **E I TP** | 1017.97 | 251.06 | 149.69 | 87.22 | 50.53 | 30.13 | 19.06 |
| **Capture Rate** | **E** I TP | 1.00 | 0.29 | 0.24 | 0.22 | 0.11 | 0.06 | 0.00 |
| | E **I** TP | 1.00 | 0.48 | 0.44 | 0.42 | 0.22 | 0.13 | 0.05 |
| | E I **TP** | 0.57 | 0.57 | 0.57 | 0.57 | 0.57 | 0.57 | 0.57 |
| | **E I TP** | 1.00 | 0.74 | 0.71 | 0.66 | 0.64 | 0.62 | 0.62 |

Table 1: Average frame counts and failure capture rate by using different thresholds and modalities

**Failure Analysis with Experience Summaries.** We systematically evaluate the capability of experience summaries generated by RONAR on failure analysis. We decompose the failulre analysis into four specific tasks: 1) Risk Estimation (Pred): if the method can identify risk before failure happens; 2) Failure Localization (Loc): if the method can identify the failure time ; 3) Failure Explanation (Exp): if the method can tell the failure reason; and 4) Recovery Recommendation (Rec): if the method can give reasonable recovery recommendations.

We compare our method with the following baseline methods and ablations (GPT-4o is used as LLM and VLM for all the methods): BLIP2, REFLECT, TEM-LLM (all raw sensory data directly input to LLM), TEM-VLM (all raw sensory data directly input to VLM), RONAR-vision (our method

uses visual inputs only), RONAR-no_prior (our method without experience history, as detailed in the Appendix). Results are shown in Figure 5.

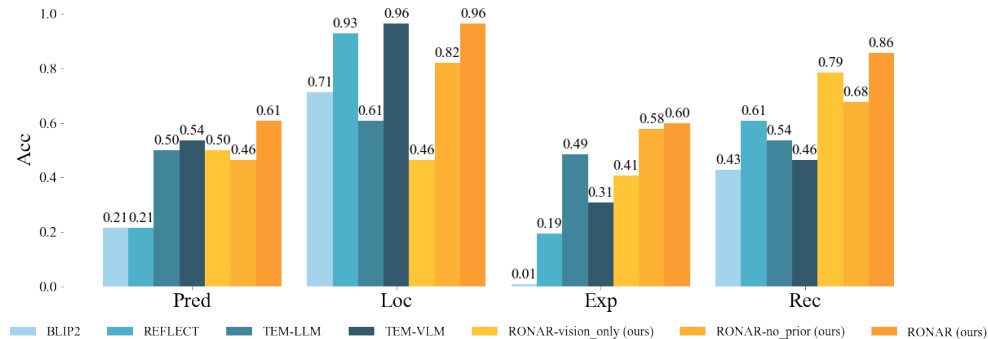

Figure 5: Accuracy on failure analysis tasks using different methods.

**User Study Setup.** We conducted a user study to evaluate our narration generation method and interface, both qualitatively and quantitatively. The user study consisted of two sections: narration quality evaluation and failure identification using narration. We recruited 24 participants for the user studies. Prior to each section, we provided a tutorial to familiarize the participants with the tasks. At the conclusion of the user study, we administered a questionnaire to gather information about the participants' demographics and their background in robotics.

**Narration Quality Evaluation (User Study 1).** Participants rated the generated narrations on naturalness, informativeness, coherence, and overall quality using a 1-5 Likert scale. The evaluation criteria for informativeness and coherence follow common practices for human evaluation of natural language generation [66]. The study commenced with an introduction to the evaluation metrics, ensuring participants fully comprehended the assessment criteria. Subsequently, participants engaged in a rating practice session involving two sample image-narration pairs to verify their understanding. Following this preparation, each participant was presented with a sequence of narrations generated by five methods, accompanied by three image-narration pairs. The results are shown in Table 2.

| Method | Naturalness | Informativeness | Coherence | Overall |
|---|---|---|---|---|
| BLIP2 | $3.25 \pm 1.18$ | $1.69 \pm 0.87$ | $2.56 \pm 1.59$ | $1.81 \pm 0.91$ |
| REFLECT | $2.13 \pm 0.95$ | $2.94 \pm 0.99$ | $2.88 \pm 1.08$ | $2.81 \pm 0.98$ |
| TEM (LLM) | $3.94 \pm 0.85$ | $4.06 \pm 1.06$ | $4.25 \pm 0.77$ | $4.13 \pm 0.71$ |
| TEM (VLM) | $3.38 \pm 0.80$ | $4.06 \pm 0.99$ | $4.06 \pm 0.85$ | $3.75 \pm 0.77$ |
| **RONAR (Ours)** | **4.19** $\pm 0.91$ | **4.56** $\pm 0.62$ | **4.56** $\pm 0.51$ | **4.50** $\pm 0.51$ |

Table 2: User ratings on narrations generated by different methods

**Failure Identification by Human with Narration (User Study 2).** We selected four failure cases in different states (two in navigation, one in manipulation, and one in detection) from the putting cup in sink task from our RoboNar dataset. We prepared four interfaces for the participants: video interface, video and sensory information interface, keyframe interface (RONAR-UI without narration), and RONAR-UI. Participants were given a full demonstration of a failure displayed on each interface. We asked participants to type their answers for the *time of failure occurrence* and *failure explanation* for each failure demonstration (see details in the Appendix). We timed participants as they answered each question using the interface. The results are shown in Figure 6.

## 4.2 Results

**RONAR can do effective failure analysis.** Our method outperforms baselines on almost all failure-related tasks, as shown in Figure 5. With a similar performance on failure localization, it achieved around 11% gain on both risk estimation and failure explanation compared with REFLECT on the

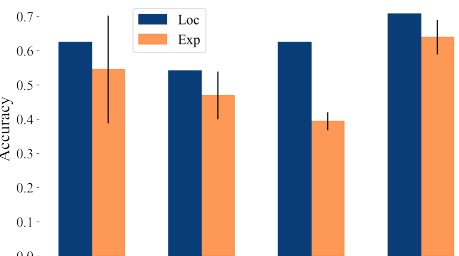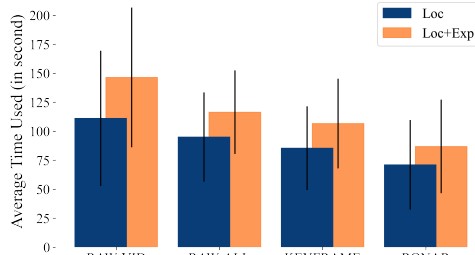

Figure 6: **Left:** The accuracy of failure localization and explanation from the user study using different interfaces. **Right:** The average time taken by participants to localize failure and both localize and explain failure while using different interfaces.

mobile robot home tasks. The result showcases the narration generated by RONAR, which can be used in robot systems to estimate risks and handle failure cases in a more effective way.

**Intermediate summarization enhances failure explanation.** As shown in Figure 5, our method has a 11% and 29% failure explanation improvements compared with the TEM-LLM method and TEM-VLM method. This result demonstrates that the LLM and VLM struggle with raw robot data from multiple sensors, whereas summarizing individual data into text first, followed by a second summarization, yields better results.

**Internal and planning data are crucial for failure analysis.** We compared variations of our methods in Figure 5. RONAR performs 50% better on failure localization and 19% better on failure explanation compared with the RONAR with vision only, showing that, in real-world scenarios, vision alone is insufficient for comprehensive failure analysis. Integrating internal and planning information enhances the accuracy of the analysis.

**RONAR can generate high-quality narrations.** In Table 2, we study the quality of the narrations on different metrics. For Naturalness, our method outperforms BLIP2, REFECLT, and TEM (VLM) with a slight improvement (0.25) on TEM (LLM). For Informativeness, all LLM-based methods have a similar performance with a big discrepancy compared with other methods. Our method has the highest overall rating, outperforming the second-place method with 0.37.

**Narration improves users' accuracy and efficiency in failure analysis.** In Figure 6, our interface outperforms other interfaces in both accuracy and efficiency in assisting users in localizing and explaining the failures. One interesting finding is that a raw data interface with all sensory inputs does not help users achieve better failure localization and explanation accuracy than a raw video interface. With similar accuracy, our method significantly reduces the time used for failure analysis compared with the raw video interface.

**Limitations.** Even with a good performance on failure analysis, some failure cases can be easily identified by humans (see Appendix). Another limitation is the latency and cost of the method. In our method, we use two-step summarization using LLM to generate narrations. This makes the system slow and affects the user experience on real robot systems. Lastly, our experiment is limited to a single mobile robot within a single environment. More types of robots and environments can be studied to the generalizability of the framework.

## 5 Conclusion

We introduce a novel framework, RONAR, which summarizes many types of diverse, raw robotic data to form an experience summary and to generate natural language narrations. The narrations can be used for improve the robot transparency and enhance failure analysis accuracy and efficiency for both robotic system and human interaction. We create a dataset for various home tasks with real failure cases and human expert labels.

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

# A    Dataset Details

## A.1    Robot Tasks and Implementations

We automated four mobile manipulation tasks in a home environment: putting a cup in the sink, microwaving food, hanging a hat, and collecting dirty clothes. The implemented tasks made up of autonomous navigation, perception, and manipulation actions. We navigate to specified positions and orientations using SLAM implemented in Stretch Nav2, detect and localize specified objects by either running ArUco marker detection or YOLO-World object detection with FastSAM segmentation, and perform manipulation using inverse kinematics with demonstrated poses. The task-specific descriptions and implementations are described below:

- **Put Cup in Sink:** The task is performed in a connected kitchen and lounge environment with a dirty cup in the lounge and a sink in the kitchen. The specific sequence of states executed by the robot is as follows: the robot navigates to a table, looks for a cup, picks up a cup from the table, navigates to the sink, looks for the sink, then places it in a sink.

- **Microwave Food:** The task is performed in a kitchen environment with a microwave and food near the microwave. The specific sequence of states executed by the robot is as follows: the robot navigates to the microwave, looks for the microwave, opens the microwave door, navigates to the food, looks for the food, picks up the food, navigates to the microwave, looks for the microwave, places the food inside the microwave, then closes the microwave door.

- **Hang Hat:** The task is performed in a lounge environment with a human wearing a hat and a hook. The specific sequence of states executed by the robot is as follows: the robot navigates to the human, is handed a hat from the human, navigates to the hook, looks for the hook, then hangs the hat on the hook.

- **Collect Dirty Clothes:** The task is performed in a lounge environment with a laundry basket and clothes arranged around the room. The specific sequence of states executed by the robot is as follows: the robot navigates to the clothes, looks around for clothes, classifies dirty clothes, picks up dirty clothes, navigates to the laundry basket, looks for the laundry basket, then places the clothes in the laundry basket.

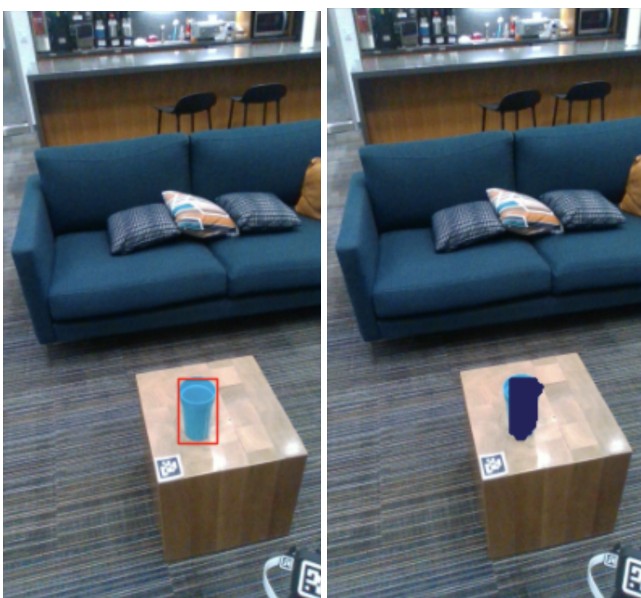

Figure 7: The robot detects and localizes the cup using YOLO-World and segments the result with FastSAM on a frame captured from its head camera for perception.

### A.1.1 State Machine Synthesis

We represent sequences of actions that accomplish a subgoal of a task as a state, and manage transitions between states using state machines. To simplify the development process, we create configurable templates to synthesize state machine code from a list of linear states. In each implemented task, we add in additional query user and teleoperation states on top of each existing action state. This allows us to manage the state of the robot after the robot experiences failures and teleoperate the robot to recover from failures.

### A.2 Dataset Overview

| Task | Trials | Failure Counts | | |
|---|---|---|---|---|
| | | Navigation | Detection | Manipulation |
| Put Cup In Sink | 32 | 10 | 2 | 18 |
| Microwave Food | 17 | 2 | 0 | 22 |
| Hang Hat | 16 | 0 | 0 | 14 |
| Collect Dirty Clothes | 5 | 0 | 5 | 0 |
| Total | 70 | 12 | 7 | 54 |

Table 3: List of autonomous tasks implemented on the Stretch mobile manipulator, along with the number of trials and failure counts separated by failure type collected for the dataset.

Tasks are run across different trials. Throughout each trial, whenever the robot encounters an incident that prevents the robot from completing its task, a failure is marked. If the failure is recoverable, a human will teleoperate the robot to resolve the failure. Otherwise, the task is aborted and marked accordingly. Room arrangements are changed to inject failures into trials. Each trial can have from zero to three failures. The tasks and failure counts are shown in Table 3.

### A.3 Data Collection

The data is collected as a rosbag that has RGB-D, joint, odometry, state information, and more beginning at the start of the task and ending after the task is complete. The full list of collected topics and topic descriptions are provided in 4.

| Topic Name | Description |
|---|---|
| /state_machine/smach/container_status | Current state status information |
| /state_machine/smach/container_structure | State machine container information |
| /camera/aligned_depth_to_color/image_raw | Head mounted D435i stereo depth |
| /camera/color/image_raw | Head mounted D435i RGB image |
| /gripper_camera/color/image_rect_raw | Gripper mounted D405 RGB image |
| /camera/aligned_depth_to_color/camera_info | Head mounted D435i camera intrinsics |
| /stretch/joint_states | Joint positions, velocities, and efforts |
| /odom | Base odometry information |
| /imu_mobile_base | Base mounted IMU information |
| /tf | Transforms for moving robot parts |
| /tf_static | Transforms for static robot parts |
| /robot_description | The robot's URDF |
| /rosout | ROS console logs |
| /diagnostics | Diagnostics information |
| /amcl_pose | The robot's Nav2 pose with covariance |

Table 4: List of all of the ROS topics and topic descriptions collected in each trial of the dataset.

# B Implementation Details

This section introduces the implementation details about the framework which is not demonstrated in the main paper due to page limit.

## B.1 Multimodal Key Event Selection

### B.1.1 A Formalism for Key Event Selection

We define a key event as the event which contains important and critical information on robot observations and robot status during task execution. These events are selected across all the modalities of the robot. Since key events can have different definitions depending on the modalities, based on the properties of the data, we categorize robot data into three categories: *Environment (E)*, *Internal (I)* and Task *Planning (TP)*:

- **Environment (E)**: Sensory data used to observe the external world, such as RGB images, point clouds, audio, tactile feedback, etc.
- **Internal (I)**: Sensory data related to the internal state of the robot, including internal sensors, joint angles, base velocity, battery levels, and other diagnostic information.
- **Task Planning (TP)**: High-level planning data that contains overall task objectives, subtask sequences, execution history, and plan outcomes.

These categorizations can group the robot data in a structured way and make key event selection across modalities easier. Then, we represent each aligned multimodal frame with the following parameters across different data categories:

- **Optical Flow (E):** the optical flows generated from the RGB images collected by the robot head camera. Since the optical flow vary largely at different stages of task operation, we define average flow magnitudes on different part movements of the robot. It has the following *parameters*:
  - $\lambda^{\mathrm{pos}}$: Average flow magnitude for frames with base positional movements.
  - $\lambda^{\mathrm{rot}}$: Average flow magnitude for frames with base rotational movements.
  - $\lambda^{\mathrm{cam}}$: Average flow magnitude for frames with camera movements.
  - $\lambda^{\mathrm{arm}}$: Average flow magnitude for frames with arm positional movements.
- **Joint State (I):** the internal joint state readings from the robot which reflects the robot internal status. It has following *parameters*:
  - $x^{\mathrm{pos}}$: Change in meters of the position of the robot from the odometry reading.
  - $x^{\mathrm{rot}}$: Change in radians of the orientation of the robot from the odometry reading.
  - $x^{\mathrm{cam}}$: Change in radians of the pan and tilt of the robot's head camera.
  - $x^{\mathrm{arm}}$: Change in meters of the robot's arm.
- **Planner State (TP)**: the planning-level state information of the robot during the task execution. It has following *parameter*:
  - $s$: The robot's current state.

For key event selection, we first calculate the mean and standard deviation of each parameter in optical flow and joint state across each task. Then, for each multimodal frame, we normalize each of these parameter values by the mean and standard deviation of the specified task. We only take multimodal frames with normalized values above 0 as the candidates of key events. The normalization on environment and internal parameters ensures values across different modalities are weighted equally, and that only values that are higher than the average frame value will be considered for contributions to the set threshold and further to become a key event. These values are accumulated along with the task execution and a set threshold is set for the cumulative sum. Additional to optical

flow and joint state, planner state is used separately for key event selection. As in task planner, the most important events are the beginning and finishing of each planner state, and therefore, these events should be also considered as key events. In summary, when either the cumulative sum of values reaches the set threshold or the planner state changes, the multimodal frame is labeled as key event. For our data, we aligned frames with a sample rate of 0.2 and set our key event threshold to 80. The key event selection can be modeled as a binary classifier, $C_{key}$, which runs across all the multimodal frames in a given task and outputs a binary prediction, 0 (not a key event) or 1 (a key event). It can be represented as following:

$$
C_{key}(f_i) = \begin{cases} 1, & \sum_{k=c}^{i} \left( \sum_{a \in \{\text{pos, rot, cam, arm}\}} (\mathcal{N}(\lambda_k^a) + \mathcal{N}(x_k^a)) \right) > \text{threshold,} \\ & \text{where } c = \text{index of last key event} \\ 1, & s_i \neq s_{i-1} \\ 0, & \text{otherwise} \end{cases} \quad (2)
$$

where $f_i$ is the multimodal frame at timestamp $i$.

### B.1.2  Adjacent Image Selection with Clarity Score

After selecting the key event, one additional step in our framework is to select the best quality RGB image adjacent to the key event. During the experiments, we found that there is a big impact on the clarity of the image for object detection, especially during navigation. Therefore, in order to improve the detection accuracy and further enhance the overall system, we define a clarity score on the RGB images and conduct an adjacent image selection. The clarity score is defined by the variance of the Laplacian of the image. The 2-D Laplacian operator is defined as:

$$
\nabla^2 f = [f(x+1, y) + f(x-1, y) + f(x, y+1) + f(x, y-1)] - 4f(x, y) \quad (3)
$$

We can use the equation to derive a $3 \times 3$ Laplacian kernel. Then, we use the kernel to convolve with the original image to get the activation map. By calculating the variance of the activation map, we can get a clarity score of the image. The higher the score is, the more clear the image is. We use the clarity score function to calculate clarity scores for all the frames within 1 second of the key event and select the RGB image with the highest clarity score as the image used for experience summarization.

### B.2  RONAR-UI

To create better user experiences, we also design an user interface for the key event and narration display, RONAR-UI. The RONAR-UI has two modes, offline mode and online mode. The offline mode is used for users to log and analyze the collected robot data. Addtionally, it is also used for user studies in C.3. The online mode is used for robot operators to lively monitor and control the robot. It integrates with

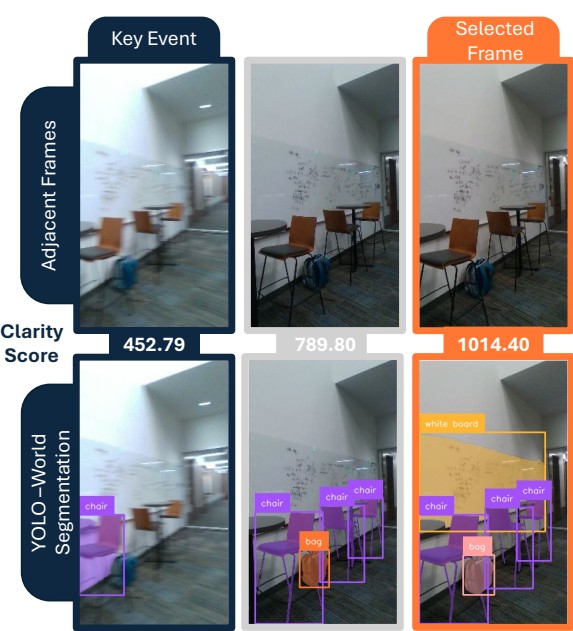

Figure 8: Adjacent image selection using clarity score. The left dark blue frame is the key event corresponding RGB image. The right orange frame is the selected frame used for experience summarization by using clarity score.

ROS2 with a user-friendly interface to allow users to learn the status of the robot system and do interventions during task execution. The RONAR-UI interfaces are shown in Figure 9.

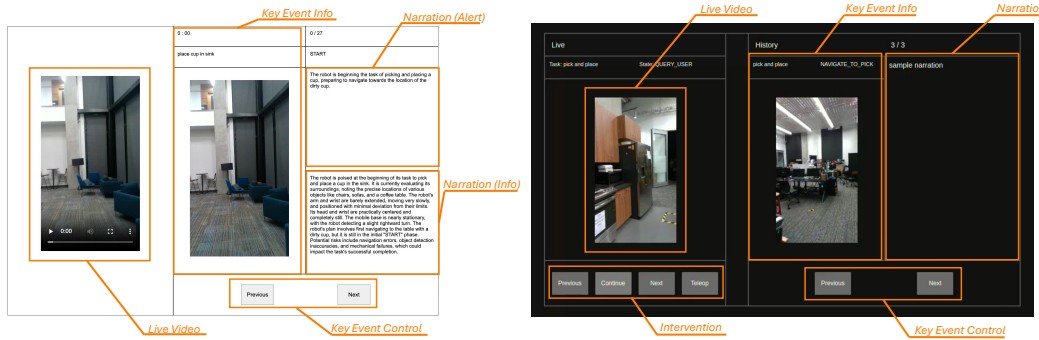

Figure 9: RONAR-UI interfaces. **Left:** RONAR-UI with offline mode. **Right:** RONAR-UI with online mode.

## C  Experiment Details

RONAR can generate multimodal experience summaries and progressive narrations in various modes. To demonstrate the effectiveness of these generated natural language groundings, we have designed several experiments. Our goal is to show that RONAR can significantly benefit two main areas:

- **Robot System:** RONAR can enhance the capability of robot systems in failure analysis.
- **Human Interaction:** RONAR can improve the interaction experience between humans and robots. Additionally, RONAR can enhance the effectiveness and efficiency of failure identification by human users.

To demonstrate the effectiveness of multimodal key frame selection, we conducted an experiment on varying modality and threshold for the key frame selection, which is shown in C.1. To demonstrate RONAR's ability to improve robot systems, we have designed an experiment focused on failure analysis using generated experience summaries, detailed in C.2. To evaluate the quality, effectiveness, and efficiency of the narrations, we have designed two user studies, described in C.3. In C.3.2, we provide details on the user study evaluating narration quality. In C.3.3, we introduce the user study focused on users' effectiveness and efficiency in identifying failures using the narrations.

### C.1  Effectiveness of multimodal Key Event Selection

The first experiment we conducted is to test the effectiveness of multimodal key event selection. The goal of key event selection is to downsample the number of frames used for narration meanwhile maintaining the information richness of the content to narrate. In order to have a fair comparison, we chose failure identification as our task and used the actual failure times in the RoboNar dataset as the ground truth for key events. We designed an additional experiment to examine the relationship between the heuristic threshold, the combination of heuristics (modalities), and the failure capture rate. The results are presented in the table below.

| | Thresh | 0 | 5 | 10 | 20 | 40 | 80 | 160 |
|---|---|---|---|---|---|---|---|---|
| **Average Frame Count** | **E** I TP | 1017.97 | 145.38 | 82.81 | 44.63 | 22.97 | 11.44 | 5.59 |
| | E **I** TP | 1017.97 | 121.72 | 68.44 | 36.97 | 19.41 | 9.75 | 4.75 |
| | E I **TP** | 11.94 | 11.94 | 11.94 | 11.94 | 11.94 | 11.94 | 11.94 |
| | **E I** TP | 1017.97 | 242.19 | 141.63 | 78.88 | 41.91 | 21.53 | 10.66 |
| | **E** I **TP** | 1017.97 | 154.25 | 91.34 | 52.84 | 31.41 | 19.53 | 14.50 |
| | E **I TP** | 1017.97 | 131.31 | 78.00 | 46.75 | 28.22 | 18.50 | 14.00 |
| | **E I TP** | 1017.97 | 251.06 | 149.69 | 87.22 | 50.53 | 30.13 | 19.06 |
| **Capture Rate** | **E** I TP | 1.00 | 0.29 | 0.24 | 0.22 | 0.11 | 0.06 | 0.00 |
| | E **I** TP | 1.00 | 0.48 | 0.44 | 0.42 | 0.22 | 0.13 | 0.05 |
| | E I **TP** | 0.57 | 0.57 | 0.57 | 0.57 | 0.57 | 0.57 | 0.57 |
| | **E I** TP | 1.00 | 0.53 | 0.49 | 0.42 | 0.32 | 0.19 | 0.03 |
| | **E** I **TP** | 1.00 | 0.72 | 0.71 | 0.67 | 0.60 | 0.59 | 0.57 |
| | E **I TP** | 1.00 | 0.71 | 0.68 | 0.65 | 0.57 | 0.57 | 0.57 |
| | **E I TP** | 1.00 | 0.74 | 0.71 | 0.66 | 0.64 | 0.62 | 0.62 |

Table 5: Average Frame Counts and Failure Capture Rate (with $\pm$ 1.5s tolerance)

As shown, there is a tradeoff between the heuristic threshold and the failure capture rate. As the threshold increases, fewer frames are sampled, which raises the probability of missing the failure frame. The selected modality also significantly impacts key event selection performance. When all modalities are enabled, fewer frames are captured, but the failure capture rate remains similar to using a lower heuristic threshold with either optical flow or joint data alone. Another interesting finding is that even with a single modality and a lower heuristic threshold (e.g. 5), achieving a

high failure capture rate is still much more sample-efficient than not using key event selection at all (captured 74.5% failures with 25% of frames).

## C.2 Failure Analysis with Experience Summary

In order to make intelligent robots, it is crucial to have reliable and efficient methods for identifying and analyzing failures. In this experiment, we aim to demonstrate that RONAR's generated experience summaries can significantly enhance the failure analysis capabilities of robot systems. To ensure a comprehensive analysis, we break down the failure analysis problem into four sub-tasks:

- **Risk Estimation / Failure Prediction (Pred):** Given previous key events, the percentage of predicted failures that are actual failures in the actual failure key event.

- **Failure Localization (Loc):** Given all key events, the percentage of predicted failure times that align with the ground truth failure times.

- **Failure Explanation (Exp):** Given previous key events and the current key event (when the failure happened), the percentage of generated failure explanations that align with the ground truth failure explanation.

- **Recovery Recommendation (Rec):** Given previous key events and the current key event (when the failure happened), the percentage of recovery recommendations that align with the ground truth recovery recommendation.

These sub-tasks cover most scenarios that robot systems encounter during operations. The methods used for comparison are:

- *BLIP2:* Uses BLIP2 to generate a caption for the RGB image of the key event.

- *REFLECT:* The current state-of-the-art LLM-based failure explanation framework.

- *TEM-LLM:* Sends all raw sensory and planning data directly to the LLM for failure analysis.

- *TEM-VLM:* Sends all raw sensory and planning data directly to the VLM for failure analysis. We use GPT-4o as the VLM.

- *RONAR-vision_only:* Our method without internal and planning inputs.

- *RONAR-no_prior:* Our method that only uses current key events for failure analysis.

Baseline methods that require an LLM use GPT-4o as the backbone. For REFLECT, we use the general pipeline of the method but do not include the audio modality since the original robot lacks a microphone, which might affect the method's performance on some tasks. The results are evaluated by human experts based on the reasonableness and deviation from the ground truth labels.

**Baseline Selection Methodology:** We selected BLIP2 and REFLECT as baselines because they represent state-of-the-art approaches in vision-language models (VLMs) and failure explanation frameworks, which are directly relevant to our task of robot experience summarization and narration. BLIP2, a strong VLM, was chosen for its ability to generate captions from visual data, allowing us to compare its performance against our multimodal summaries. REFLECT, designed for failure explanation in robotic systems, serves as a benchmark for how well our system explains failures. Additionally, we included LLM-TEM and VLM-TEM to establish baseline capabilities for summarizing raw robotic data using large language models, highlighting the improvements our method offers through key event selection and progressive narration. RONAR-Vision_Only isolates the performance of our system using only visual data, demonstrating the necessity of multimodal integration. Finally, RONAR-No_Prior was selected to assess the impact of progressive narration generation, illustrating the benefits of using prior narrations for more coherent summaries.

### C.3 User Studies

In order to prove the quality, effectiveness and efficiency of RONAR narrations, we carefully design two experiments, narration quality evaluation and failure identification using narration, and recruit a number of participants to conduct a thorough user study.

#### C.3.1 Demographics and Robot Background of Participants

We recruited 24 participants with different background to conduct the user studies on narration quality evaluation and failure identification using narration. We create a questionnaires to ask for participants' background at the end of the user study. The questionnaires include two parts: demographics and robot background. For the demographics, we include the following questions:

- **Age:** Select your age range (under 18, 18-24, 25-35, 35-44, 45-54, 55-64, over 65)
- **Education:** What is your highest level of education? (High school diploma / GED, Associate degree, Bachelor's degree, Master's degree, Doctorate)
- **Filed:** Have you studied or worked in a tech or STEM related field? (Yes or No)

The details of participant demographics can be found in Figure 10. From Figure 10, we can see that the user study involved a predominantly young and highly educated group of participants. The age distribution shows a significant concentration in the 18-24 age range, while the educational background highlights that most participants hold at least a Bachelor's degree, with nearly half having attained a Master's degree. This demographic profile suggests that the findings of the study might be particularly relevant to younger, well-educated individuals.

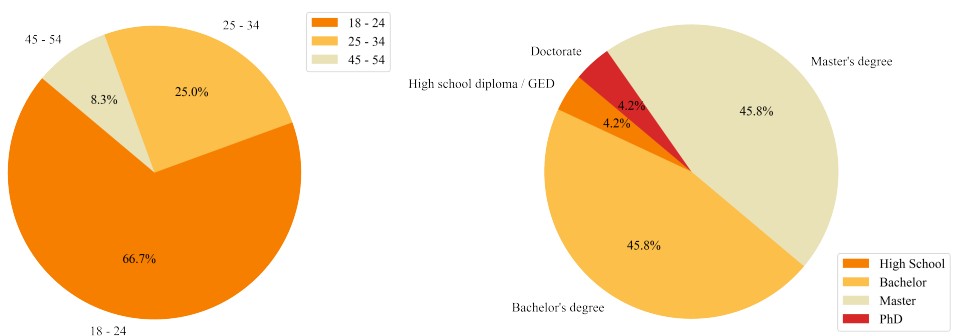

Figure 10: Demographics of the participants. **Left:** Age ranges of the participants of the user study. **Right:** Highest degrees earned by the participants of the user study.

We also create questions on robot familiarity for participants to answer. These questions inlcude:

- **Expertise Level (Robot):** Rate your expertise in robotics (1-5)
- **Hours Spent (Robot):** Estimate how many hours have you worked with real robot? (Never, 0-10h, 10-30h, 30-50h, 50-100h, More than 100h)
- **Expertise Level (Stretch):** Rate your expertise with Hello Robot's Stretch mobile manipulator (1-5)
- **Hours Spent (Stretch):** Estimate how many hours have you worked with stretch robot? (Never, 0-10h, 10-30h, 30-50h, 50-100h, More than 100h)

The details of participant expertise on robots can be found in Figure 11. From Figure 11, it is clear that while the participant pool is quite diverse in terms of general robotics experience, they predominantly lack familiarity with the Stretch robot. Despite a few individuals with substantial robotic experience, the overall expertise with Stretch is low. Furthermore, the distribution of expertise level

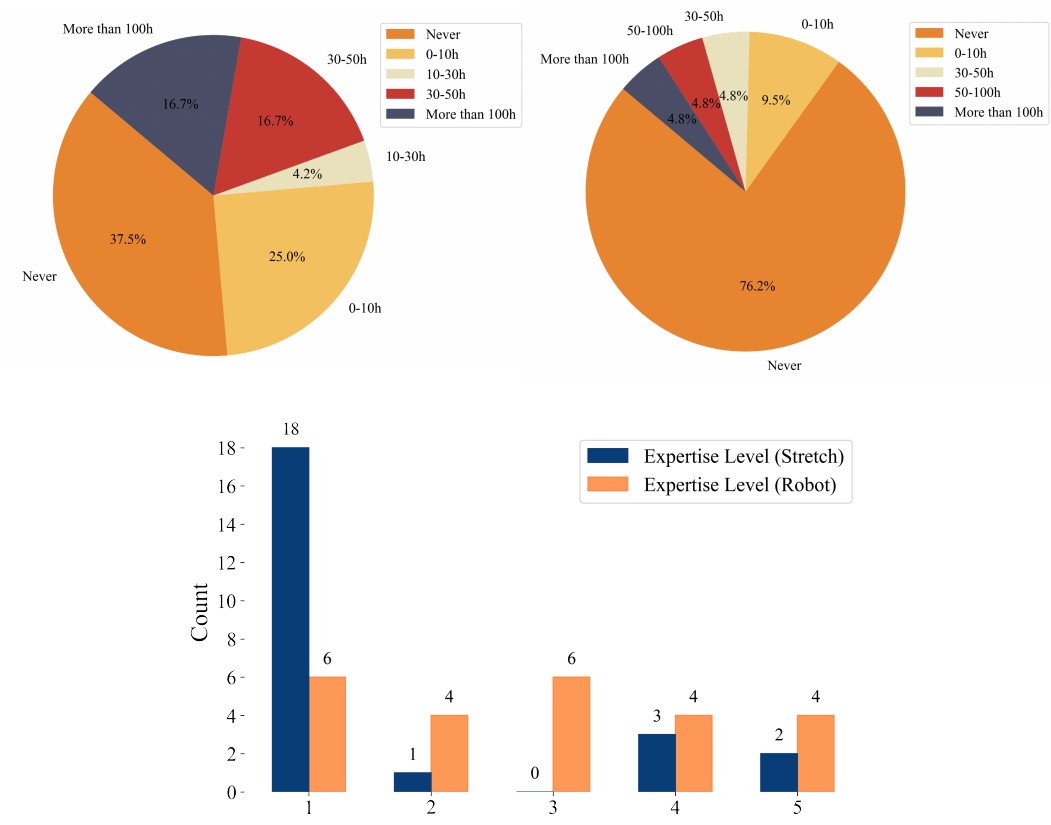

Figure 11: Expertise of the participants with robot and Stretch. **Top Left:** Hours spent by the participants on robots in general. **Top Right:** Hours spent by the participants on Stretch. **Bottom:** Subjective self-evaluation on the expertise level of the participants on general robot and Stretch.

on general robots shows a relatively even spread of expertise levels, with a notable concentration at both the novice and intermediate levels, reflecting a varied participant pool in terms of general robotics knowledge.

### C.3.2 Narration Quality Evaluation

The study of narration quality evaluation has two parts: a tutorial and a formal evaluation. Before the start of the tutorial, we introduce the four metrics the participants give ratings on:

- **Naturalness:** Does the narration feel natural and human-like? (1-5)
- **Informativeness:** Does the narration provide useful information about the robot's behavior? (1-5)
- **Coherence:** Does the narration organize information logically and clearly? (1-5)
- **Overall:** What is your overall assessment of the narration's quality? (1-5)

It uses 1 to 5 Likert Scale to measure the preferences from the participants on the metrics. We confirm and explain the metrics until the participants fully understand the task and the terminologies. Then, the participants are given a short tutorial on two samples of the image-narration pairs to get familiar with the format and questions. They can ask any related questions during the tutorial. After the tutorial, it goes to the formal evaluation of the narrations generated by different methods. In the formal evaluation, we select three frames from three tasks in the dataset: put cup in sink, microwave lunch and hang hat. There are 5 methods evaluated by the participants: BLIP2, REFECLET, TEM-

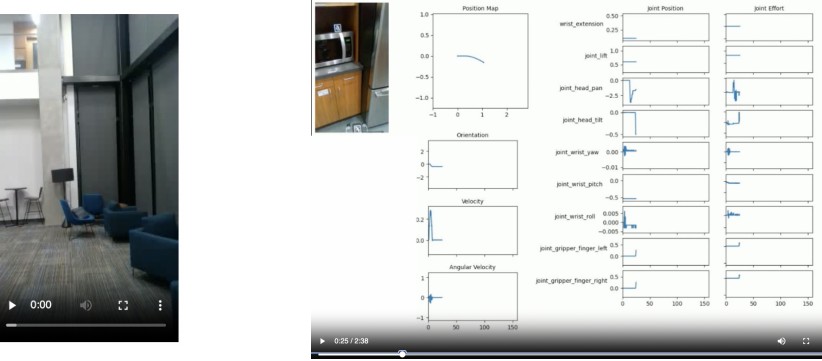

(a) RAW-VID interface            (b) RAW-ALL interface

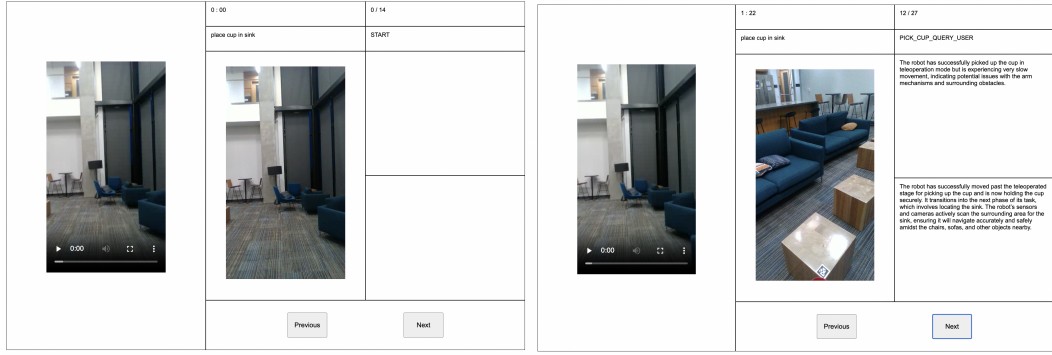

(c) KEYFRAME interface          (d) RONAR-UI interface

Figure 12: Four interfaces used for failure identification in user study.

LLM, TEM-VLM and RONAR. We pair the three selected frames with the corresponding narrations generated by different methods. In order to prevent biases from the orders of methods, for each participant, we generate a random order of methods and all the names of the methods are hidden. After the participants saw all three image-narration pairs, they can give scores on the four metrics for the corresponding method. Since the order is very critical and users might change their mind by seeing the following narrations, we allow the users to go back and change their ratings by seeing the new narrations.

### C.3.3 Failure Identification Using Narration

The second user study is to evaluate the effectiveness and efficiency of narrations on failure identification. The key question we want to answer is how effective and efficient the narration can help users to identify the failures. We design two tasks for the failure identification problem:

- **Failure time identification:** How accurate and efficient can participant correctly identify the time of failure in a demo?
- **Failure explanation:** How accurate and efficient can participant have a reasonable explanation about the failure?

In this study, we design four interfaces which the users will be used to identify the failure time and failure reason. The four failure identifications interfaces are:

- **RAW-VID:** a traditional video player interface which only shows the raw video captured by the robot camera.
- **RAW-ALL:** a video player interface which displays the raw video and all raw sensor readings (both joint and base) from the robot. The sensor readings are visualized by line plots and synchronized with the raw video.

- **KEYFRAME:** the RONAR-UI interface without narration. It includes the raw video, selected keyframes, and state information.
- **RONAR-UI:** the RONAR-UI interface with fully functionalities. It includes both narrations with alert mode and info mode generated RONAR.

The appearances of all interfaces can be found in Figure 12. The effectiveness and efficiency are measured by *accuracy* and *time spent* on each of the task. For failure time identification, we have the ground truth failure time and set a time tolerance for each of the demonstrations. If the participant gives an estimated time within the time tolerance, it is marked as correct. Otherwise, it is marked as incorrect. For failure explanation, we ask the participants to write 1-2 sentences to explain the failure in each of the demonstration. Then, three robot experts evaluate the users' answer based on reasonableness independently. The accuracy on failure explanation is the average of rate of correctness marked by the experts. For measuring efficiency, we record the time which participants used to fulfill each task. This time is measured by an expert supervisor with a stopwatch beside the participant. For failure time identification, the start time is when the user click the play button of the video and the finish time is when the user finish typing and click *next* button. For failure explanation, the start time is when the user saw the ground truth failure time and the finish time is when the user finish typing and click *next*.

This study is also consist of two parts: an introductory tutorial and a thorough evaluation of interfaces. In the tutorial, we prepare examples of each interface for users to interact with. We give enough time for them until they are familiar with all the interfaces and comfortable to continue for the actual evaluation. For the formal evaluation, we select four demonstrations from the task, *put cup in sink*, with failures in different states and we make sure that each demonstration only contains one failure. These are the failures for the formal evaluation. In order to prevent the biases from the demo-interface pairs, we make a full set of permutations of the four demos and four interfaces, which results 24 permutations. We assign each different permutation to different participants, which covers exact 24 participants.

# D Experience Summary and Narration Examples

## D.1 Experience Summary Example

**Environment Summary**

**I observe:**
*stool_0:* 0.93 meters and in front of the robot, left of sofa_0, right of coffee table_0.
*sofa_0:* 1.13 meters and in front of the robot, right of chair_0, left of chair_1, right of stool_0, left of coffee table_0.
*chair_1:* 1.24 meters and above the robot, right of chair_0, left of sofa_0, left of stool_0.
*chair_0:* 1.25 meters and above the robot, left of sofa_0, left of stool_0, right of coffee table_0.
*coffee table_0:* 1.27 meters and in front of the robot, right of chair_0, right of chair_1, left of sofa_0, right of stool_0.

**Internal Summary**

**Here is what I am doing:**
*wrist_extension:*
  *Descriptions:* position at 19.23% of its maximum extension, velocity nearly at a standstill with a very slight retraction (-0.000226 m/s), effort is very minimal (4.39623047631315e-45%).
  *Grounded:* My arm is mostly retracted and not moving much.
*joint_lift:*
  *Descriptions:* position at 54.59% of its maximum height, velocity nearly at a standstill with a very slight downward motion (-0.000206 m/s), effort at 40.91% of maximum torque.
  *Grounded:* My arm is raised to about halfway up and holding steady.
*joint_head_pan:*
  *Descriptions:* position at 76.49% of its maximum left pan, velocity at standstill (0.0 rad/s), effort at 0.0%.
  *Grounded:* My camera is facing mostly straight ahead and not moving.
*joint_head_tilt:*
  *Descriptions:* position at 64.78% of its maximum upward tilt, velocity at standstill (0.0 rad/s), effort at 0.0%.
  *Grounded:* My camera is tilted slightly upwards and not moving.
*joint_wrist_yaw:*
  *Descriptions:* position at 15.97% of its maximum yaw to the right, velocity at standstill (0.0 rad/s), effort at 0.0%.
  *Grounded:* My wrist is slightly rotated to the right and not moving.
*joint_wrist_pitch:*
  *Descriptions:* position at -94.22% of its maximum downward pitch, velocity at 0.0 rad/s, effort at -0.10%.
  *Grounded:* My wrist is pitched downward quite a bit and holding steady.
*joint_wrist_roll:*
  *Descriptions:* position at 50% of its range, velocity at 0.0 rad/s, effort at 0.0%.
  *Grounded:* My wrist is in a neutral roll position and not moving.
*joint_gripper_finger_left:*
  *Descriptions:* position at 50% of its opening range, velocity at 0.0 rad/s, effort at 0.0%.
  *Grounded:* The left gripper finger is halfway open and not moving.
*joint_gripper_finger_right:*
  *Descriptions:* position at 50% of its opening range, velocity at 0.0 rad/s, effort at 0.0%.
  *Grounded:* The right gripper finger is halfway open and not moving.
*position:*
  *Descriptions:* x at 1.627e-5 m, y at -4.526e-11 m.
  *Grounded:* My base is almost at the same spot, with very minimal movement.
*orientation:*
  *Descriptions:* orientation at -0.0000289 radians.
  *Grounded:* I am nearly in the same direction I started.
*velocity:*
  *Descriptions:* velocity moving forward at 0.000166 m/s.
  *Grounded:* I am barely moving forward.
*angular_velocity:*
  *Descriptions:* angular velocity rotating clockwise at -0.00187 rad/s.
  *Grounded:* I am turning right very slowly."



**Planning Summary**

**My plan is:**
***Task:*** Pick and place the cup in the sink:
   Description: The robot should navigate to the table with dirty cup on it, pick up the cup, find the sink, navigate to the sink and place the cup in the sink.
***Overall plan:***
   Step 1: Begin operation from the START state.
   Step 2: Navigate to the location of the cup.
   Step 3: Execute pick operation to grab the cup.
   Step 4: Navigate to the location of the sink.
   Step 5: Detect the precise position to place the cup.
   Step 6: Place the cup in the sink.
   Step 7: End the operation and transition to END state.
***Current state:*** NAVIGATE_TO_CUP
***State History:*** [['START', 'success']]



## D.2  Narration Examples

We have shown narration examples generated by RONAR with different modes in the main paper. In this section, we want to show some interesting cases the narration can help users to identify the status of the robot and failures the robot is experiencing.

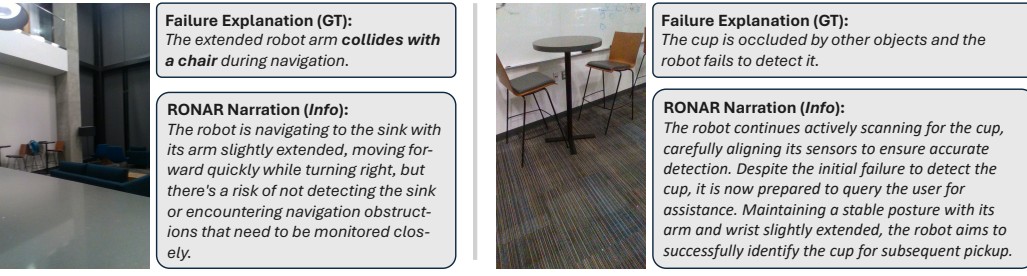

**Failure Explanation (GT):**
*The extended robot arm **collides with a chair** during navigation.*

**RONAR Narration (*Info*):**
*The robot is navigating to the sink with its arm slightly extended, moving forward quickly while turning right, but there's a risk of not detecting the sink or encountering navigation obstructions that need to be monitored closely.*

**Failure Explanation (GT):**
*The cup is occluded by other objects and the robot fails to detect it.*

**RONAR Narration (*Info*):**
*The robot continues actively scanning for the cup, carefully aligning its sensors to ensure accurate detection. Despite the initial failure to detect the cup, it is now prepared to query the user for assistance. Maintaining a stable posture with its arm and wrist slightly extended, the robot aims to successfully identify the cup for subsequent pickup.*

Figure 13: Narration examples generated by RONAR.

As shown in Figure 13, the left example shows an interesting failure case which cannot be identified by the vision system alone. In the left example, the robot arm is extended and the arm hits a chair which impedes the movement of the robot. As shown in the frame, users cannot identify what the failures the robot is current facing. During the user study, most participants reason the failure is caused by the robot cannot successfully find and locate the sink, which is due to the detection and mapping errors. The RONAR narration successfully captures the robot arm status (*"its arm slightly extended"*) and the movement of the robot (*"moving quickly while turning right"*). As well, it notifies the users of the potential risks which could cause failures. It successfully identifies the risks of navigation obstruction during the task execution without direct visual information.

In the right example, the robot fails to detect the cup with the first attempt. From the visual inputs, it is hard to identify what the robot is doing. Most users consider the robot is navigating to the sink even by watching the video. The narration generated by RONAR successfully captures the past experiences of the robot and explains clearly what the robot is currently doing. In summary, demonstrated by the user studies, it shows significant improvements for users on understanding robot behaviors by using the narrations generated by RONAR.

Furthermore, a comparison of narrations generated by different baseline methods and our method can been seen in Figure 14. In this example, BLIP provides a very general description, lacking any specific reference to the robot's actions or the challenges it faces. REFLECT offers more detail than BLIP but focuses heavily on the static description of objects and the robot's position, without much insight into the robot's task progress or the difficulties encountered. LLM-TEM provides a

sequential summary of actions but lacks depth in describing the current status of the task and the specifics of the failure. Similar to LLM-TEM, VLM-TEM offers a basic description of the robot's actions but does not effectively convey the ongoing challenges or next steps. RONAR provides a comprehensive, real-time narrative of the robot's current state, the specific issue it is encountering, and the intended corrective action. This level of detail allows users to understand both the problem and the proposed solution, making RONAR's output significantly more actionable and informative.

This example provided demonstrate that RONAR excels in offering detailed, contextually rich, and actionable narratives compared to the baselines. RONAR not only describes what the robot is doing but also offers insight into challenges and next steps, which are crucial for effective monitoring and intervention. This depth of analysis and real-time relevance is why RONAR consistently outperforms the baseline models.

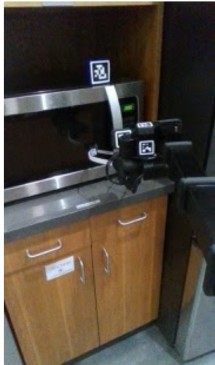

*BLIP: a microwave sitting on top of a kitchen counter*

*REFLECT: The robot, equipped with a camera, is positioned such that it is above cabinet_0 and to the right of microwave_0, and also above cabinet_1. From this camera angle, it observes that cabinet_0 is situated directly below the microwave_0 and to the right of another cabinet, cabinet_1, which is also below the microwave. The robot's immediate goal is to open microwave_0.*

*LLM-TEM: The robot has successfully navigated and positioned itself to open the microwave, with its camera oriented to accurately view the microwave above a cabinet. Currently, the robot's joints are actively engaged, particularly its gripper fingers, as it performs the task of opening the microwave door. It has completed the initial stages of its task sequence as reflected in the state history, advancing through the start, navigating, and detecting stages to reach the microwave successfully. The robot efficiently maintains its orientation and velocity as it operates within the environment.*

*VLM-LLM: The robot has successfully navigated to the microwave and opened it, as indicated by the successful completion of the 'NAVIGATE_TO_OPEN_MICROWAVE' and 'DETECT_TO_OPEN_MICROWAVE' stages. The robot's joints, particularly the grippers and wrists, are positioned to interact with the microwave, which is consistent with its task of opening the microwave. The odometer readings indicate precise positioning near the microwave, suited for further interactions like placing food inside. The visual observation confirms the robot's gripper is in close proximity to the now-opened microwave, ready for the next steps in its task.*

*RONAR: The robot is currently in the process of opening the microwave. Observing its environment, there are no other objects detected nearby, ensuring a clear path. The robot's arm is mostly extended and raised high, poised for the next step, while the grip fingers are halfway open and slowly closing. The robot has successfully navigated to the microwave and detected its handle or opening mechanism. The next step is to manipulate the microwave door to open it.*

Figure 14: Comparison of Narrations Generated by Different Methods.

## D.3 Extensions of Narration

The use of narrations uss not only limited to failure analysis and transparency. There could be much more applications and extensions. We study some of the extensions and welcome for more follow-up research on related studies.

### D.3.1 Trajectory Summarization

The narration generated by RONAR is event-level, meaning each narration corresponds to a single key event. It captures a snapshot of the process but cannot provide an overview of the entire demonstration. To address this, we create a higher-level summarization, called trajectory summarization, generated by LLMs using the narration history. This summarization captures the details of the trajectory in a human-readable way.

Trajectory summarization enables the question-and-answer capability of robot trajectories and creates richer ways of interacting with robot data. One potential use of trajectory summarization is for customized trajectory retrieval. As shown in

**Trajectory Summary**

The robot began its task of picking and placing a cup in the sink at **0:00**, initially assessing an environment cluttered with obstacles such as stools and chairs. After finalizing its initial assessment by **0:02**, it navigated towards the cup on the table, carefully maneuvering around furniture. By **0:13**, upon reaching the table, the robot transitioned to a state where it scanned for the cup before successfully identifying and picking it up. After identifying the cup by **0:23**, the robot transitioned to pick up the cup, secure the cup and prepare to move towards the sink. Next, the robot navigated towards the sink, moving cautiously while continually adjusting for minor obstacles and ensuring the cup remained secure by **0:43**. Despite successful navigation, the robot initially failed to correctly place the cup into the sink by **1:45**, prompting user intervention. Following user guidance, the robot retried and successfully maneuvered into position for precise placement. By **2:21**, the robot's careful adjustments in arm and wrist movements aimed to ensure the cup was accurately positioned before final release. The task finished with the robot successfully placing the cup into the sink.

Figure 15: Trajectory summarization

Figure 16, for a collection of robot demonstrations, RONAR can generate a corresponding trajectory summary for each trajectory. These summaries can then be used for trajectory retrieval purposes. Users can search for and retrieve trajectories with customized queries. These queried trajectories can be used for model training in various types of learning algorithms, such as imitation learning. This approach makes robot data search and retrieval much more efficient and accessible.

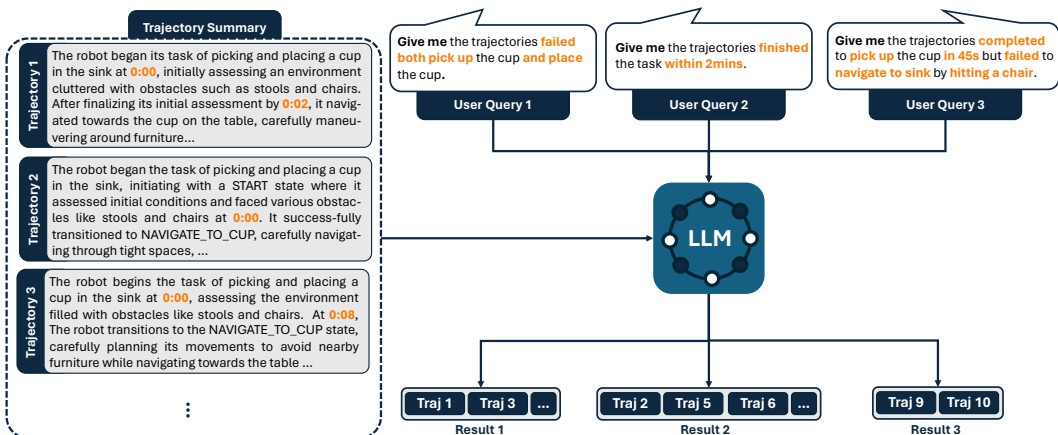

Figure 16: **A pipeline for customized trajectory retrieval.** For each demonstration, RONAR can generate trajectory-level summarization by using event-level narrations. This summarization contains the detailed information of the trajectory and users can retrieve trajectories with customized queries. These customized trajectories can be used for further downstream tasks, such as model training and system analysis.

### D.3.2 System Overview

Not limited to trajectory-level summaries, RONAR can generate summaries in an even higher-level. With a collection of trajectory summaries, RONAR can generate a system-level summary to give users an overview of the robot system. As shown in Figure 17, users can generate robot system overview by using a collection of trajecory summaries and RONAR. The system overview is customizable based on users' requirements. In this example, we ask RONAR to generate system overview on failures, recoveries and improvement recommendations based on the experiments. The system overview can give users a big picture of the overall system and assist to make improvements. As well, users can also compare system overviews between different experiment dates to keep track of the system improvement progress.

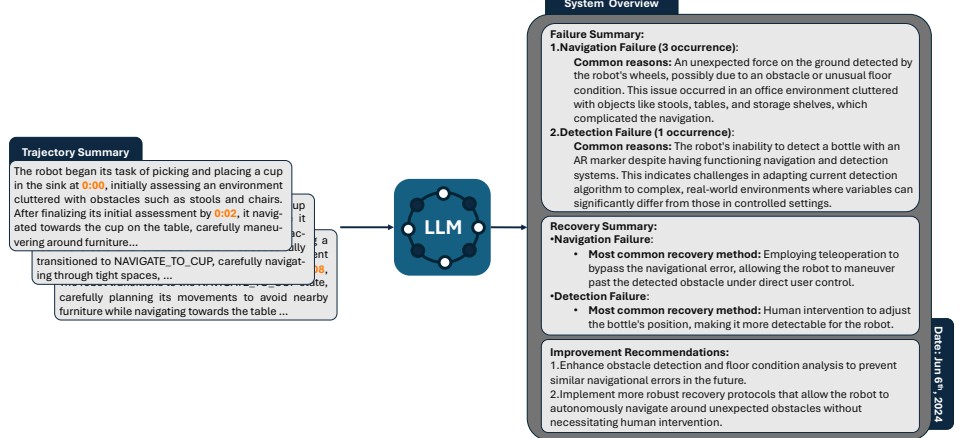

Figure 17: System overview generated by RONAR

