# OpenReview forum: "I Can Tell What I am Doing: Toward Real-World Natural Language Grounding of Robot Experiences"
_robot-learning.org/CoRL/2024/Conference — CoRL 2024_

### Official Review · Reviewer_kuGW · 2024-07-14
**Limited novelty and evaluation**

**Originality:** 2
**Technical Quality:** 2
**Clarity Of Presentation:** 2
**Potential Impact:** 3
**Recommendation:** 3
**Confidence:** 4

**Review:**

This paper addresses the important problem of generating natural language descriptions of robot experiences to improve transparency and failure analysis. The overall idea of using large language models to narrate robot data is interesting. However, there are several major weaknesses that limit the impact and novelty of this work.

Strengths:
1. Addressing an important problem: Improving robot transparency and failure analysis through natural language is a valuable direction.

2. User study: The inclusion of a user study provides some evidence of real-world utility.

3. Multiple narration modes: The different narration modes (alert, info, debug) are a nice feature to support different use cases.

Weaknesses:
1. Limited technical novelty: The core framework appears to be a relatively straightforward application of GPT-4 to summarized robot data. The paper does not present significant algorithmic or methodological innovations beyond this.

2. Lack of clarity on baselines: The results section compares to methods like BLIP2 and REFLECT without clearly explaining why these are appropriate baselines or how they were adapted to this task. This makes it difficult to interpret the comparative results.

3. Small-scale evaluation: The dataset of only 70 demonstrations and 76 failure cases is quite limited. This raises questions about the generalizability of the results and whether they support broad conclusions about the method's effectiveness.

4. Insufficient analysis: The paper does not provide adequate insight into why RONAR outperforms baselines or analysis of failure cases. More in-depth examination of example outputs would strengthen the evaluation.

**Quality Of The Limitations Section:**

1

**Questions For Rebuttal:**

1. Can you provide more details on the technical innovations of RONAR beyond applying GPT-4 to summarized robot data?

2. Why were BLIP2, REFLECT, etc. chosen as baselines? How were they adapted to this task?

3. How generalizable are the results given the limited dataset size? What steps were taken to ensure diversity of scenarios?

4. Can you provide a more in-depth analysis of example outputs and failure cases?

===post rebuttal
Rebuttal addressed most of my questions. I still think the small dataset and technical novelty limit the impact of your work. However, I have upgraded my rating to weak accept.

**Robotics Focus:**

4

**Summary Of Paper:**

This paper presents RONAR, a framework for generating natural language narrations of robot experiences using large language models. The system takes in multimodal robot data, summarizes it into text, and uses GPT-4 to generate narrations in different modes (alert, info, debug). The authors create a dataset of home robot tasks with failure cases and conduct experiments on failure analysis and user studies. They claim RONAR improves transparency and failure analysis compared to baselines.

**Summary Of Recommendation:**

I recommend a weak accept for this paper. While the overall direction is promising, the limited technical novelty, small-scale evaluation, and lack of clarity in the results section significantly weaken the contribution. The straightforward application of GPT-4 to summarized robot data, without substantial algorithmic innovations, limits the depth of the work. The paper would need major extensions and a more comprehensive evaluation to be suitable for publication at CoRL.

---

### Official Review · Reviewer_6w4H · 2024-07-19
**Interesting approach for robot behavior narration**

**Originality:** 3
**Technical Quality:** 3
**Clarity Of Presentation:** 4
**Potential Impact:** 3
**Recommendation:** 3
**Confidence:** 4

**Review:**

Strengths
- The presented method is interesting and addresses and important issues of transparency of robot behaviors that can help with debugging robotic policies or ensuring explainability and safety of robotic systems.
- The paper is well-written and easy to understand and follow.
- The experiments show that the produced narrations explain well the robotic application in a variety of real-world scenarios.

Criticism
- As the method uses various additional components to process raw data, it would be useful to see an extended discussion on how failures or mistakes in these components affect the narration. For example, robot planning can come up with a wrong plan or the mapping from the sub-goals to robot actions might be sub-optimal. In this case, robot movements would not be coherent with the current plan or task goal, which might also lead to sub-optimal narration.
- As the system gets a lot of information from components that extract human-readable information, it would be interesting to understand if the information could be directly summarized without LLMs, e.g. by directly implementing text templates to construct a human-readable text.
- It would be interesting to see a discussion on how the system could be further improved through fine-tuning the VLM model, e.g. on human expert narrations.

**Quality Of The Limitations Section:**

2

**Questions For Rebuttal:**

Please refer to review for questions.

**Robotics Focus:**

4

**Summary Of Paper:**

The paper presents a framework for narrating robot behaviors based on a range of multi-modal sources of information. In particular, the paper proposes several components for processing such information as: environment object information (using an object detector) and robot camera images, robot parts and robot state (explained in plain language including numerical values), the summary of the current task, plan and sub-goals including detection of keyframes for important events. All the processed information is fed into a large VLM/LLM such as GPT-4o to output descriptions of what is happening in the scene and possible failure modes. In addition, paper proposes different levels of narration detailization such as alert, info and debug modes. The experiments show that the framework can successfully produce coherent narrations and outperforms previous methods that mostly directly consume raw sensory data.

**Summary Of Recommendation:**

The presented framework is effective at narration, however uses several components that already extract human-readable information that either might fail on their own leading to the full system failure, or be used directly without LLMs to output human-readable text.

---

### Official Review · Reviewer_7Xr6 · 2024-07-20
**Evaluation of key event selection and narration quality**

**Originality:** 3
**Technical Quality:** 3
**Clarity Of Presentation:** 3
**Potential Impact:** 3
**Recommendation:** 3
**Confidence:** 4

**Review:**

Post-rebuttal comments

I would like to thank the authors for their detailed rebuttal, which clarified the design decisions of the key event selection, UI, and evaluation.

---

The proposed RONAR framework contributes to human-robot interaction by increasing the transparency of robots to users with natural language key event narration. The three modes of information abstraction (alert, info, debug) potentially allow diverse users to customise the presented narration based on their robot expertise and interaction goal. The effectiveness of the proposed framework was evaluated using empirical experiments with different task scenarios and failure cases.

This work can be further improved by clarifying the design of heuristics-based key event identification and extending the evaluation, as detailed below.

1. Key event selection

As discussed in Sections 3.1 and B.1, key events were selected when either the sum of optical flow and joint state heuristics surpassed a pre-defined threshold, or when there was a state change in the task planner. It would be interesting to understand the alignment between key events selected by each of these three heuristics (optical flow, robot motion, state change), the unions of each pair and all three, the intersections between each pair and all three, and a groundtruth annotation of key events. As the narration is generated from these selected key events, the accuracy and completeness of key event selection will influence the quality of the resulting narration and the failure analysis based on it.

2. Evaluation of narration quality

As described in Sections 4.1 and C.2.2, a custom questionnaire was used to evaluate the naturalness, informativeness, coherence, and overall quality of the generated narration. Since the goal of the narration is to increase a user's understanding of the robot, especially on failure analysis, this narration may be considered an explanation. Thus, existing questionnaires in the XAI literature for evaluating the quality of explanations may be applicable here to extend the measure on narration quality.

In the user study evaluation, participants were given image-narration pairs to aid their failure analysis. Both alert and info mode narrations were provided in the RONAR-UI, which matched the participant population being largely novice or having intermediate expertise. However, presenting both narrations may provide overloaded information. It may be interesting to allow participants to select from the three modes (alert, info, debug) and analyse the frequency of each mode being used by participants with different expertise level. Further, the choice of which narration mode to use may also relate to the task context, such as the concise alert mode being preferred for risk estimation or time-critical failure recovery, while the info and debug mode being preferred for in-depth analysis of past failures or developing new failure prevention mechanisms.

Regarding failure identification using narration, as described in Section C.2.3, the failure analysis time was measured by an observer with a stopwatch and the finish time was defined as when a user finished typing their answer to failure time and cause. It may be more accurate to measure this time automatically using button / keyboard events captured in the UI, and to set the finish time as when the user started typing their answer to not disadvantage long answers.

**Quality Of The Limitations Section:**

3

**Questions For Rebuttal:**

- The project website (https://sites.google.com/view/real-world-robot-narration/home) appears to be offline
- Please clarify performance of the multimodal key event selection
- Please clarify the rationale of presenting both alert and info mode narrations in the RONAR-UI and how the different narration modes may influence participants' understanding of the robot and its failures.

**Robotics Focus:**

4

**Summary Of Paper:**

This paper presents a LLM-based robot narration generation framework RONAR. User study evaluation demonstrated RONAR's benefits in supporting end users to understand robot failures.

**Summary Of Recommendation:**

While the evaluation of this work can be extended to analyse the multimodal key event selection and the narration quality in more depth, the proposed framework contributes to increasing end users' understanding of robots and their failures. Thus, I recommend this paper to be accepted.

---

### Author Rebuttal · Authors · 2024-08-12

We first want to thank all the reviewers for their insightful and constructive feedback on our manuscript. We appreciate the time and effort taken to provide detailed comments, and we are sure that the revisions made in response to these comments have significantly strengthened our work.

We have replied to each reviewer's comment individually in the official comment section, providing specific responses and outlining the changes and clarifications made to address each point.

Here, we want to give some general comments on the major revisions and modifications in our *manuscript*, *supplementary material* and *website* (Note: all the changes made in the manuscript and supplementary material are using **blue** color):
- **Effects of Heuristic Combinations and Heuristic Threshold for Key Frame Selection:** Based on the reviewers' suggestions, we have conducted additional experiments to explore the effects of different heuristic combinations and heuristic thresholds on key frame selection. The results of these experiments are now included in both the main *manuscript* and the *supplementary material*. We have provided a detailed analysis of how varying the heuristic thresholds impacts the key frame selection performance.
- **Detailed Justification of Baseline Selection:** We provided a thorough explanation of why specific baselines, such as BLIP2 and REFLECT, were chosen and how they were adapted to our task. This justification is included in the *supplementary material*, clarifying the relevance and appropriateness of these comparisons to our method.
- **Additional Examples and Analysis of Baseline vs. RONAR:** To further clarify the performance differences between our proposed method (RONAR) and the selected baselines, we have added examples and comparative analysis in the *supplementary material*. These additional examples, along with detailed analysis, highlight the strengths and weaknesses of each approach. We have also made these examples available on our *project website*.

We believe these revisions address the key concerns raised by the reviewers and enhance the overall contribution of our work. We are grateful for the feedback and hope that the revisions meet the expectations of the reviewers.

We sincerely thank the reviewers once again for their valuable comments and recommendations. Please feel free to reach out if there are any further questions or if we can provide additional clarification!

---

### Decision · Program_Chairs · 2024-09-04

**Decision:**

Accept

**Comment:**

This paper addresses an interesting problem of describing the robot experience in natural language derived from low-level percepts. The paper provides a mechanism to narrate behaviors, failures and human interaction to recover from failures. Reviewers have suggested a better exposition of heuristic-design and insights into why this approach improves over the baselines. The authors provided a detailed response during the rebuttal phase. In particular, additional results on heuristic thresholds vs. failure capture rate, UI details, key event selection and further discussion on effects of modality failures and mistakes on narration generation. The reviewers are in consensus for a weak accept for this paper.